# Difference in All-Cause Mortality between Unemployed and Employed Black Men: Analysis Using the National Health and Nutrition Examination Survey (NHANES) III

**DOI:** 10.3390/ijerph20021594

**Published:** 2023-01-16

**Authors:** Paul Delgado, Dulcie Kermah, Paul Archibald, Mopileola T. Adewumi, Caryn N. Bell, Roland J. Thorpe

**Affiliations:** 1Office of Medical Student Research, OSU College of Osteopathic Medicine, Tulsa, OK 74107, USA; 2Hopkins Center for Health Disparities Solutions, Johns Hopkins Bloomberg School of Public Health, Baltimore, MD 21205, USA; 3Urban Health Institute Student Research Core Charles R., Charles R. Drew University of Medicine and Science, Los Angeles, CA 90059, USA; 4Department of Social Work, College of Staten Island, City University of New York, Staten Island, NY 10314, USA; 5Department of Social, Behavioral, and Population Sciences, Tulane University School of Public Health and Tropical Medicine, New Orleans, LA 70112, USA

**Keywords:** mortality, employment, unemployment, Black men

## Abstract

The Black-White racial employment disparity and its link to mortality have demonstrated the health benefits obtained from employment. Further, racial/ethnic mortality disparities existing among men with different employment statuses have been previously documented. The purpose of this study was to examine the association between employment status and all-cause mortality among Black men. Data for the study was obtained from the National Health and Nutrition Examination Survey (NHANES) III 1988–1994 linked to the NHANES III Linked Mortality File. Cox proportional hazard models were specified to examine the association between health behaviors and mortality in Black men by employment status. Among those who were assumed alive (n = 1354), 41.9% were unemployed. In the fully adjusted model, unemployed Black men had an increased risk of all-cause mortality (Hazard Ratio [HR] 1.60, 95% confidence interval or CI [1.33, 1.92]) compared to Black men who were employed. These results highlight the impact of employment on all-cause mortality among unemployed Black men and underscore the need to address employment inequalities to reduce the mortality disparities among Black men.

## 1. Introduction

According to the United States Department of Labor, the US unemployment rate was 6.1% in April of 2021 [1]. The unemployment rate among Black Americans (9.7%) was almost twice as high as the unemployment rate among White Americans (5.3%) (4). Over the past few decades, the study of unemployment and its association with health suggests that unemployment is associated with negative health outcomes and all-cause mortality [2]. Unemployment has often been associated with poor health through multiple factors including stress, depression, loss of social networks, increased substance use, self-harm, and a reduction of self-efficacy further leading to unhealthy behaviors and poorer decisions about medical care [2,3]. For example, unemployed individuals have a 2.46 higher risk of dying from opioid overdose compared to those that are employed [4]. Research-based on an intersectional approach suggests that health gain associated with employment is conditional on one’s race, gender, and education level [5]. 

There is limited literature on the link between race, employment, and all-cause mortality [3,5]. More specifically, there is a lack of research examining the impact of employment on all-cause mortality among Black men. Previous research on the Black-White racial employment disparity and its link to mortality demonstrates that we must address the way the labor market operates to address health disparities [5,6]. While the US unemployment rate has dropped over time, Black adults have lower overall earnings growth accompanied by declining health in analyses of differences in employment status in the Black-White earnings gap [6,7]. There is a paucity of research addressing disparities that emerge among Black adults. Moreover, attention should be given to Black men in particular given the large racial inequities in unemployment between Black and White men [8].

The passage of the Civil Rights Act and the Equal Employment Opportunity Commission in the 1960s made significant contributions to upward employment mobility for Black men [9]. The gap in hourly wages and the racial gap in male employment rates have continued throughout the decades [8]. Barriers identified by Black men resulting in unemployment include traditional forms of discrimination based on demographics, to new forms of discrimination based on credit score, criminal records, and appearance [8]. In addition, the association between employment discrimination and mortality may have a greater impact on groups that experience higher rates of unemployment, such as Black men [10]. Estimates of life expectancy based on employment status suggest that a 25-year-old, employed non-Hispanic Black men, could expect to live 14.0 more years, on average, compared to those not in the labor force (i.e., persons engaged in their housework, those unable to work, sick, and disabled) [11]. Therefore, job market opportunities and equal access to employment, equal pay, paid sick leave, and unemployment benefits, are key foundations for life expectancy gain among Black men [5,12,13]

The Black-White disparity between mortality and employment continues to widen with Black men experiencing persistent shortfalls. Although substantial literature [8,12,14] has explored the impact of employment on racial health disparities, there is limited research on the relationship between employment status and all-cause mortality among Black men. The objective of this study is to examine the association between employment status and all-cause mortality among Black men. Specifically, we anticipated unemployed Black men to have an increased risk of all-cause mortality than employed Black men. The findings from our study make an important contribution to the existing literature examining employment and extend to understanding the relationship among Black men, further underscoring the need to address employment inequalities.

## 2. Materials and Methods

### 2.1. Survey Design and Data Collection 

The study sample was obtained from the National Health and Nutrition Examination Survey (NHANES) III 1988–1994. NHANES III is a cross-sectional survey conducted by the National Center for Health Statistics that uses a stratified multistage probability sample to obtain a representative sample of the total US population [15]. The data obtained from NHANES is comprised of information collected during interviews conducted at home, clinical information during an examination or via a questionnaire, and data obtained from blood samples. Our study is exempted/waived from informed consent as we used de-identified data available for public use. Our analytic sample consisted of 2300 African American men 20 years and older. 

### 2.2. Outcome Variable

The Third National Health and Nutrition Examination Survey (NHANES III) is a nationally representative sample of the civilian noninstitutionalized population (total n = 39 695) with an oversample of African Americans (n = 11,061) and Mexican Americans (n = 11,110), children younger than 5 years, and adults aged 60 years and older (n = 2273). In addition to the NHANES III publicly available data, we used data from the NHANES III Linked Mortality Public-Use File that contains limited death certificate data. This file provides the opportunity for mortality follow-up on the NHANES III participants through 31 December 2006 [16]. We utilized data from the NHANES III Linked Mortality File to estimate non-injury related death rates for NHANES III participants using a probabilistic matching algorithm, linked to the National Death Index through 31 December 2006 [16] and provided up to 18 years of follow-up (mean [SE] 14 [0.2] years).

### 2.3. Exposure Variable

Employment status, our exposure variable, was based on the question “During the past 2 weeks, did you work at any time at a job or business, not counting work around the house?” A binary variable was created to identify those African American men who answered “Yes” as Being Employed, and those African American men who answered “No”, as Unemployed.

### 2.4. Covariates

We included the following covariates: age, education, income, marital status, representing social, behavioral, and health-related factors. Demographic variables included age measured in years. Education was categorized into three groups based on participants’ number of years of education: less than high school, high school/GED, some college, and above. Income was based on the poverty-income ratio (PIR) of less than two versus a PIR of two or greater. The cut points were selected based on census poverty and income indicators. Marital status was categorized into married, widowed/divorced/separated/living with a partner, and never married. Health insurance was coded as to whether participants had insurance or not. Smoking status was categorized into current, former, and never smokers based on the question “Do you smoke cigarettes now?”. Alcohol use was based on consumption of alcohol per day during the past month and categorized into non-drinker, 1–30 drinks/month, >30 drinks per month. Self-rated health was categorized into fair/poor, good/very good/excellent based on the question, “How would you rate your overall health?” [17,18].

### 2.5. Statistical Analyses

Weighted frequencies and means were used to summarize the demographic and health-related characteristics by employment status. Chi-square and ANOVA tests were employed to examine proportional and mean differences by employment status among Black men. Cox proportional hazard models were specified to examine the association between employment status and all-cause mortality among Black men. Four models were specified. Model 1 adjusted for age; Model 2 added SES (education, poverty-income ratio, health insurance status) and marital status; Model 3 added health behaviors (smoking and drinking status) to Model 2; Model 4 added self-rated health indicators to Model 3. In the results section, results are expressed as hazard ratios (HRs) with 95% confidence intervals (CIs). Survey procedures were used when specifying the statistical models to account for the complex survey sampling design of NHANES III. All analyses were performed using SAS v 9.4 (Research Triangle Park, NC, USA). *p* values < 0.05 were considered statistically significant and all tests were two-sided.

## 3. Results

The distribution of the select baseline characteristics of Black men in NHANES III for the total sample and by employment status is presented in Table 1. Unemployed Black men were more likely to have less than a high school education (28.7%, n = 311), more likely to have a PIR of <2 (77.6%, n = 610), be a non-drinker (44.7%, n = 431) and more likely to have fair/poor health status (37.0%, n = 509) compared to employed Black men. In addition, unemployed Black men were less likely to have health insurance coverage (78.5%, n = 688), be married (42.6%, n = 396), less likely to report never smoking (30.7%, n = 246), and less likely to be assumed alive (41.9, n = 284) compared to employed Black men.

The association between employment status and all-cause mortality is presented in Table 2. Across all four models, there was an increased risk of all-cause mortality for Black men who were unemployed compared to Black men who were employed. In the fully adjusted model which accounted for age, education, poverty-income ratio, health insurance status, marital status, smoking status, alcohol use, and self-rated health, unemployed Black men had an increased risk of all-cause mortality (Hazard Ratio [HR] 1.60, 95% confidence interval or CI [1.33, 1.92]) compared to Black men who were employed.

## 4. Discussion

The purpose of this study was to examine the association between employment status and all-cause mortality among Black men. Our studies indicate that unemployed Black men have an increased risk of all-cause mortality compared to unemployed Black men. These results highlight the impact of employment status on all-cause mortality among Black men. Further, this work underscores the need to address employment inequalities to reduce all-cause mortality among Black men.

The findings from this study indicate significant differences in all-cause mortality between employed and unemployed Black men. This work is consistent with prior scholarship focusing on the association between employment status mortality among working-aged men [6,7,9]. However, our work extends to understanding the relationship among Black men using a nationally representative sample. There are several key explanations for our findings in the context of Black men that may contribute to the differences based on employment status.

As expected, this is a complex issue with several factors such as education, access to health insurance, lifestyle choices, and many more playing a role as discussed above. Taking into consideration the fundamental structure of American society, these findings are quite formulaic. American culture has historically favored the influential, affluent, members of the majority race, men, and generally people with higher levels of education. Several studies have previously highlighted the direct correlation between higher socioeconomic status (SES), higher education, and better health outcomes [19,20]. On a sub-population level, this does not necessarily hold. The Black’s diminished return suggests that Blacks have a smaller return of SES and are more likely to develop poor health outcomes despite an increase in the level of education when compared with their white counterpart [21]. Considering the correlation between education and employment it is thus expected that this diminishing return hypothesis may hold within subpopulations of Black men. This is supported by a study by Bell, Sacks, Tobin, & Thorpe (2020) that suggested a diminished return of education on health among Black men but not for Black women. Other areas of inquiry to explain health disparities, such as discrimination, psychosocial factors, and segregation were proposed [22]. 

Following Bell and colleagues’ (2020) line of inquiry, we examine our results through the lens of discrimination and segregation. The relationship between discrimination and mortality has been examined over time showing positive correlations [20,23,24]. This is important because the employment discrimination experiences of Black adults have remained constant since 1990 [25]. The effects of employment discrimination may have a greater impact on Black men since they have unemployment rates higher than any other race/gender group [10]. After controlling for differences based on skills and occupational differences, the “last in, first out” phenomenon associated with discriminatory employment practices, suggests that there is strong evidence that employment practices vary according to race [26]. Given discrimination’s association with mortality, exposure to employment discrimination may be affecting the rates of mortality among unemployed Black men. The study by Gilmore, Whitfield, & Thorpe (2019), suggests the importance of examining the level of education when examining all-cause mortality. Our findings suggest that a higher level of education has the potential to make it easier to find a job and thus plays a role in other factors such as access to health insurance, having a higher PIR, and ultimately better health status.

When considering segregation, Black men tend to live in neighborhoods that are experiencing forced racial residential segregation further decreasing employment opportunities [20]. Racial residential segregation separates the affluent from the poor and produces unequal space between job demand and job availability which contributes to high unemployment in neighborhoods with higher concentrations of low incomes [27]. Racially segregated neighborhoods oftentimes have access to fewer resources to assist with adequate housing, safe neighborhoods, quality education, and healthcare, as well as other resources that have been identified to influence health status [28,29]. In this context, racial residential segregation may serve as a surrogate of stress-related morbidity; exacerbated by the limited stress-buffering resources available to unemployed Black males exposed to forced racial segregation in neighborhoods, which in turn, places them at a higher risk for mortality.

Our study makes an empirical contribution to the existing body of research examining employment status and all-cause mortality. However, it does have some noteworthy limitations including a population sample collected between 1988–1994. While the NHANES III is based on a random representative non-institutionalized sample of the US population, study participants may differ from those who are not participating and may have an impact on study results and the generalizability of the findings. It is also important to mention that a cross-sectional study cannot define cause-and-effect relationships. A major limitation of our study is the classification of employment status based on the question “During the past 2 weeks, did you work at any time at a job or business, not counting work around the house?”. We recognize this artifact as a limitation of our study as the variable was created based on “Yes” as Being Employed, and “No”, as Unemployed in NHANES III. One last critical limitation of our study is the use of a proportional hazard model with time-invariant covariates as it may have influenced our model estimation. The choice of covariate selection should be considered for future studies. However, despite these limitations, we are unaware of any study that focuses on employment status and all-cause mortality among Black men.

Our work is focused solely on employment status and all-cause mortality in Black men and future work should consider exploring all-cause mortality based on specific causes of death.

## 5. Conclusions

Employment status is a key social determinant of health for Black men and a key foundation for potentially improving health outcomes. In this study, we found that there is an increased risk of all-cause mortality in unemployed Black men compared to unemployed Black men. These findings underscore the need to address employment inequalities. Further studies are warranted to explore in detail the nuances of unemployment that may be catalyzing mortality among Black men.

## Figures and Tables

**Table 1 ijerph-20-01594-t001:** Distribution of Select Baseline characteristics of Black men in the National Health and Nutrition Examination Survey III for the total sample and by employment status (n = 2300).

	Total n (%)	Unemployed n (%)	Employed n (%)	*p* Value
**Age Mean (SE)**	41 (0.4)	37 (0.3)	50 (0.8)	
**Education**				<0.0001
Less than High School	443 (13.8)	311 (28.7)	132 (6.9)	
High School Graduate/GED	1235 (56.1)	428 (56.4)	807 (56.0)	
Some College and Above	586 (30.1)	108 (14.8)	478 (37.1)	
**Poverty-Income Ratio (PIR) < 2**	1187 (53.1)	610 (77.6)	577 (42.0)	<0.0001
**Insurance Coverage**	1850 (83.0)	688 (78.5)	1162 (85.0)	<0.0001
**Marital Status**				<0.0001
Married	1245 (53.0)	396 (42.6)	849 (57.8)	
Widowed/Divorced/Separated	438 (17.9)	248 (26.0)	190 (14.2)	
Never married	604 (29.1)	216 (31.5)	388 (28.1)	
**Smoking Status**				<0.0001
Never	844 (40.0)	246 (30.7)	598 (43.5)	
Former	539 (20.7)	250 (23.8)	289 (19.3)	
Current	913 (39.8)	369 (45.5)	544 (37.2)	
**Drinking Status**				<0.0001
Non-drinker	925 (36.4)	431 (44.7)	494 (32.6)	
<1 alcoholic drink/day	1085 (50.6)	333 (31.8)	752 (54.7)	
>1 alcoholic drink/day	285 (13.0)	100 (13.5)	185 (12.7)	
**Fair/Poor Health Status**	1753 (80.3)	509 (37.0)	1244 (11.7)	<0.0001
**Assumed Alive**	1354 (67.1)	284 (41.9)	1070 (78.7)	<0.0001

**Table 2 ijerph-20-01594-t002:** Association between all-cause mortality and employment status for Black men in National Health and Nutrition Examination Survey III (n = 2300).

	Adjusted			
	Model 1	Model 2	Model 3	Model 4
**Unemployed**	1.94 (1.69–2.23)	1.71 (1.42–2.07)	1.70 (1.42–2.03)	1.60 (1.33–1.92)

Model 1 adjusts for age; Model 2 adds education, poverty-income ratio, health insurance status, and marital status to the covariates in Model 1; Model 3 adds smoking and drinking status to the covariates in Model 2; Model 4 adds self-rated health to the covariates in Model 3.

## Data Availability

Publicly available datasets were analyzed in this study. The data can be found at https://wwwn.cdc.gov/nchs/nhanes/nhanes3/default.aspx and https://www.cdc.gov/nchs/data-linkage/mortality-public.htm (accessed on 4 September 2021).

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
