# Peer review of "Difference in All-Cause Mortality between Unemployed and Employed Black Men: Analysis Using the National Health and Nutrition Examination Survey (NHANES) III"

_ijerph, 2023, doi:10.3390/ijerph20021594_

Round 1
Reviewer 1 Report
Summary: This paper studies the impact of unemployment on racial mortality disparities in the US. In particular, the relationship between employment and mortality experience among Black men is examined. The data used is obtained from the NHANES III. It is found that unemployment can contribute to a higher level of mortality for Black men.
Comments: The introduction section produces a very interesting opening to the research question, and also a concise review on the existing studies. I would recommend a quick summary on the contributions and findings of this paper in the introduction as well.
The data used comes from a survey over 1988–1994. I wonder whether using a more recent dataset will change or exacerbate the conclusion of this paper. This could be just a discission or future work.
I think the authors can more specifically describe the four Cox models used. For example, what does “Model 1 adjusted for age” means? Also, for a general redear, how to interpret the important statistics Hazard Ratios could be explained before reaching a conclusion from Table 2.
Reviewer 2 Report
Dear authors,
The study is based on surveys collected for a different purpose 35 years ago. I think this study is not well designed. In addition, the following is noted:
INTRODUCTION
Authors should include some epidemiological data to support their claims. The data quoted for 2016 are not current. Moreover, there is no mortality data to support the authors' hypothesis.
MATERIALS AND METHODS
2.1. Survey design and data collection:
The analysis is based on a population sample collected between 1988-1994. The US population has changed a lot since then. I think the authors should use a more up-to-date sample.
.2. Outcome Variable:
The authors have compared data from two different surveys. If the subjects are not linked, you cannot infer conclusions from analysed data related to each other.
DISCUSSION
Limitations
The authors say that "It is also hard to denote or establish causality based on the cross-sectional nature". This is not correct. It is not that it is difficult, it is that causal relationships cannot be established in a cross-sectional study.
The information is from 1988-1994. There have been many global events that may have influenced lifestyle habits and therefore the causes of death: economic crises, climate crisis, pandemics...
CONCLUSIONS
The conclusions are bold. No causal relationships can be established by methodological design.
REFERENCES
Many bibliographies are obsolete. The bibliographic citations used are more than 5 years old (39.4 %). The authors must update and arrange the bibliography
Round 2
Reviewer 2 Report
Dear authors,
Thank you very much for attending to the suggestions and comments. The article has improved in quality. Congratulations for your work.
Best regards